# DIRECTIONALITY IN GRAPH TRANSFORMERS

## ABSTRACT

We study how one can capture directionality in graph transformers, for learning over directed graphs. Most existing graph transformers do not take edge direction into account. We therefore introduce a novel graph transformer architecture that explicitly takes into account the edge directionality. To achieve this, we make use of dual encodings to represent both potential roles, i.e., source or target, of each pair of vertices linked by a directed edge. These dual encodings are learned by leveraging the latent adjacency information extracted from a novel directional attention module, localized with $k$-hop neighborhood information. We also study alternative approaches to incorporating directionality into other graph transformers to enhance their performance on directed graph learning tasks. To evaluate the importance of edge direction, we empirically characterize via randomization whether direction really matters for the downstream task. We propose two new directional graph datasets where direction is intrinsically related to learning. Via experiments on directional graph datasets, we show that our approach yields state-of-the-art results.

## 1 INTRODUCTION

Graphs are one of the most general and versatile data structures that are encountered in diverse application domains, ranging from biology and social networks to transportation and finance. Analyzing the graphs that arise from such applications and discovering patterns in them is of paramount importance in the associated domains. An important property of a graph is whether its edges are directed or not. Directed graphs are natural representations of relations including social connections, human communications, paper citations, financial transactions, web links, and causes and effects. The state-of-the-art methods for analyzing directed graphs use Graph Neural Networks (GNNs) to learn node and directed edge encodings for tasks like link prediction (Kollias et al., 2022; Salha et al., 2019), node classification (Zhang et al., 2021) and graph-level tasks (Beaini et al., 2021).

In this paper, we address the relatively unexplored problem of analyzing *directed* graphs using graph transformers (GTs). Transformers hold the promise of enhanced performance over GNNs due to their ability to represent entities without enforcing the inductive adjacency bias (Vaswani et al., 2017), and due to their *dynamic* multi-head attention mechanism, in contrast to GNNs where the attention is hardwired in *static* edge weights. This flexibility of GTs comes, however, with the challenge of modeling directed graph structures. Most existing GT works focus on integrating only the graph connectivity structure into the Transformer. They do not prioritize how to reflect the directionality of graph edges in their proposed architecture. This is either due to the fact that one of their key techniques is not applicable to directed graphs (e.g., Laplacian eigenvectors (Dwivedi & Bresson, 2021)) or the edge-direction information is encoded as static, fixed scalars (either local in/out degrees in (Ying et al., 2021) or pairwise shortest path distances in (Hussain et al., 2022; Ying et al., 2021)).

We introduce *Directed Graph Transformer* (DiGT), a novel GT architecture that explicitly takes into account graph directionality. The crux of this architecture is that it incorporates both edge direction and graph connectivity structure into the standard Transformer architecture (Vaswani et al., 2017) as first-class citizens. Edge direction is represented by dual encodings for each graph node capturing its potential role as either a source or target of a directed edge. A node encoding in DiGT thus consists of a pair of source and target vectors. Edge direction information is preserved in node representations: the attention between a query node $i$ and a key node $j$ will yield different values depending on whether $i$ points to $j$, or $j$ points to $i$. Thus, the node encodings produced by DiGT

embed edge direction semantics in them. These source and target encodings are then learned using a multi-head *directional attention* module that incorporates edge channels as bias. By interpreting attention matrices as latent adjacency matrices, our technique updates a node's source vector by aggregating the target vectors of the neighbors it points to, after incorporating suitable learnable parameters; similarly, a node's target vector update is the aggregation of the source vectors of those neighbors pointing to it. It is important to note that in DiGT dual node encodings are *dynamically* learned without using the explicit directed graph structure, whereas previous approaches exploited the static connectivity information only.

Our main contributions are as follows: i) We propose a new directed graph Transformer architecture (DiGT) that uses dual node encoding approach with source and target encodings, and a novel directional attention mechanism, ii) We propose several alternative strategies for incorporating direction into graph transformers, in terms of exploiting single or dual attention matrices, and we show that these approaches can be incorporated into different GT architectures like (vanilla) Transformers (Vaswani et al., 2017), EGT (Hussain et al., 2022), and Exphormer (Shirzad et al., 2023), to improve their performance on directed graphs, iii) We characterize the "directionality" of a graph dataset, showing that some of the popular directed graph benchmarks like `MNIST`/`CIFAR10` (Dwivedi et al., 2020), `Ogbg-Code2` (Hu et al., 2020) are not truly directional. Therefore, we introduce the `FlowGraph` and `Twitter` family of directed graph datasets that explicitly relate the edge direction pattern in graphs to their classification labels. iv) We compare our proposed DiGT model against other GNN and GT variants for directed graph classification tasks. Our experiments reveal that when edge directionality is an inherent, rather than derivative, characteristic of the instances to be classified, DiGT can beat the best state-of-the-art (SOTA) GT and GNN alternatives by a large margin.

## 2 RELATED WORK

**Methods for Directed Graph Learning.** Earlier works on analyzing directed graphs are based on matrix factorization techniques to learn node encodings, such as Singular Value Decomposition (SVD) of higher-order adjacency matrices exploring the directed $k$-hop neighborhood of a node (Ou et al., 2016), or Non-negative Matrix Factorization (NMF) (Sun et al., 2019). Another line of work focuses on analyzing special matrix forms of adjacency information such as the Hermitian adjacency matrix of the directed graph (Cucuringu et al., 2020) or learning linear combinations of powers of the directed graph adjacency matrix and its transpose (He et al., 2021). APP (Zhou et al., 2017) uses random walks with restart as a tool to scale to large graphs and harvests asymmetric and high-order similarities between node pairs. More recently, GNNs have been used (Kollias et al., 2022; Tong et al., 2020a;b; Zhang et al., 2021). Compared to the above techniques, GNNs operate based on a message-passing architecture and provide higher learning flexibility due to the usage of learnable weight matrices that multiply the node encodings. Graph Attention Network (GAT) (Veličković et al., 2017) is a GNN that incorporates local self-attention resembling a transformer.

A limitation of all the aforementioned approaches is that they critically rely on the explicit directed graph structure (adjacency matrix): (a) The $k$-hop neighborhood learning techniques (He et al., 2021; Ou et al., 2016; Sun et al., 2019; Zhou et al., 2017) involve matrix factorization or composition of powers of known adjacency matrices, or random walks over the graph structure. (b) Special matrix forms of adjacency information used in (Cucuringu et al., 2020; Tong et al., 2020a) require the directed graph as input. (c) GNNs in (Kollias et al., 2022; Salha et al., 2019; Tong et al., 2020b; Veličković et al., 2017; Zhang et al., 2021) are message-passing models, and single or dual-node encoding messages can flow only through existing edges. Reliance on the directed graph structure introduces inductive bias during learning: latent edges that could positively contribute to the learning problem at hand can be missed as a result. In contrast, DiGT does not rely on the directed graph structure (in effect it assumes full graph connectivity) and learns the edge weights by exchanging the dual node encodings between the nodes.

We note here that the general idea of dual node encodings has also been used in several of the above works. However, such encodings are typically computed by exploiting the directed graph structure. Applying this idea in GTs is challenging because there are no assumptions on graph structure (i.e. assuming full connectivity) and they need to be learned in a dynamic manner.

**Graph Transformers** GTs were introduced in (Dwivedi & Bresson, 2021), which proposed two inspiring GT architecture variants. The first variant produces only node encodings, while the second variant is augmented to also produce edge encodings. Node encodings follow the standard Transformer architecture (Vaswani et al., 2017), while edge encodings are updated by scaling the attention matrix. They attend only to existing neighbors (local self-attention), so a strong inductive bias is enforced. SAN (Kreuzer et al., 2021) uses learned positional encodings (LPE) to enhance the learning of graph structure. SAT Chen et al. (2022) enhances the learning by extracting k-hop subgraphs. In Graphormer (Ying et al., 2021), the attention aperture critically expands to all nodes (global self-attention). They propose adding and learning node encodings that are functions of input and output degree centralities (centrality encoding), and arbitrary node pairs are represented by two bias terms to the attention matrix (spatial and edge encodings). In Edge-Augmented Transformer (EGT) (Hussain et al., 2022), they combine ideas from (Dwivedi & Bresson, 2021) (separate channels for nodes and edges, scaling and gating the attention matrix) and from (Ying et al., 2021) (global self-attention, bias terms from spatial encoding, however, learned from the edge channels) to yield an effective GT approach. Recently, the models that combine graph neural networks with graph transformers, such as GraphGPS (Rampášek et al., 2022) and Exphormer (Shirzad et al., 2023), attained competitive performance results, while aiming at scalability. GraphGPS is a framework for combining pluggable encoding, local message passing, and global attention modules; Exphormer introduces a sparse attention mechanism based on global virtual nodes and expander graphs. In (Geisler et al., 2023), they learn directed graphs using Transformers which leverage special positional encodings based on Magnetic Laplacian eigenvectors and random walks.

Our DiGT approach is a global self-attention transformer, learning both dual node encodings and edge encodings (dual-channel architecture). A node encoding in DiGT consists of a pair of source and target vectors that capture the edge direction semantics. Therefore, in downstream tasks that require directionality and take node encodings as input, DiGT provides embeddings of high discriminative power. In comparison, Graphormer (Ying et al., 2021), EGT (Hussain et al., 2022), and the other graph transformers (Dwivedi & Bresson, 2021; Zhang et al., 2020) produce only single-vector node embeddings that cannot differentiate the direction of an edge; and, as already mentioned, directed GNNs (Kollias et al., 2022; Salha et al., 2019; Tong et al., 2020b; Veličković et al., 2017; Zhang et al., 2021) produce dual node embeddings that suffer from convolutional inductive bias, that is restricted to only the given neighborhood structure.

## 3 DiGT: DIRECTED GRAPH TRANSFORMER

We now describe our DiGT directed graph transformer that uses three main ideas: dual node embeddings for source and target representations; which are combined with learnable implicit adjacency information via directed attention; as well as using $k$-hop neighborhood information. We will detail these ideas below. Our model typically contains multiple DiGT layers, as well as multiple heads for the attention. However, in the description below, we omit the layer and head notations for ease of presentation.

### 3.1 INPUT LAYER

We represent a directed *graph* as $G(V, E)$; $V$ is the set of $n = |V|$ graph nodes, $E = \{(i, j) \in V \times V : i \mapsto j\}$ is the set of its $m = |E|$ directed edges. Each node $i$ is equipped with a pair of vectors in $\mathbb{R}^d$, $1 \le i \le n$: (i) vector $\mathbf{s}_i$ encodes $i$'s role as a source, which is the same for any of the directed edges it participates in as a source, and (ii) vector $\mathbf{t}_i$ encodes $i$'s role as a target.

Given the $n \times n$ adjacency matrix $\mathbf{A}$ of the input (directed) graph $G$, consider its truncated SVD, $\mathbf{A} \sim \mathbf{U}_r \boldsymbol{\Sigma}_r \mathbf{V}_r^\top$, where we keep the $r$ largest singular value triplets, and let us set $\mathbf{S}_r = \mathbf{U}_r \boldsymbol{\Sigma}_r^{\frac{1}{2}}$ and $\mathbf{T}_r = \mathbf{V}_r \boldsymbol{\Sigma}_r^{\frac{1}{2}}$ for the source and target positional encodings. When input node features $\mathbf{X}_f$ are available (set $\mathbf{X}_f = 0$ otherwise), the input/initial *node* embeddings for the DiGT model are given as

$$\mathbf{S} = L_s(\mathbf{S}_r) + L_f(\mathbf{X}_f) \qquad\qquad \mathbf{T} = L_t(\mathbf{T}_r) + L_f(\mathbf{X}_f) \qquad\qquad (1)$$

where $L_s$, $L_t$ and $L_f$ are learnable linear transformations (subscripted as $s$ for the sources, $t$ for targets, and $f$ for input features), and $\mathbf{S}, \mathbf{T} \in \mathbb{R}^{n \times d}$.

For encoding the edges, if input edge features $\mathbf{E}_f$ are available (set $\mathbf{E}_f = 0$ otherwise) the input/initial *edge* embeddings for the DiGT model are given as

$$\mathbf{E}_{ST} = L_e([\delta_{st}]_{s,t=1,\dots,n}) + L_{ef}(\mathbf{E}_f) \tag{2}$$

where $L_e$ is an embedding layer, $L_{ef}$ is a learnable linear transformation, and $\delta_{st}$ is the shortest directed path distance from source $s$ to target $t$, clipped at maximum $k$-hops (if $t$ is not reachable from $s$ we set $\delta_{st} = k + 1$). The result, $E_{ST} \in \mathbb{R}^{n \times n \times d_e}$, is the matrix of $d_e$ dimensional edge embeddings, and we set $\mathbf{E}_{TS}$ as the transpose of $\mathbf{E}_{ST}$ along the first two dimensions.

## 3.2 DiGT Attention Layer

Given the dual node encodings, we need to determine the relationship between the source and target encoding vectors of different nodes, which will be used for updates in our GT architecture. For this, we draw high-level inspiration from the HITS centrality algorithm (Kleinberg, 1999) that computes two scalar-valued *hub* and *authority* scores for each node in a directed graph – a source node with a high hub score refers to (or points to) target nodes that contribute relevant information (in our case, for learning), and thus gain elevated authority scores. Consider for the moment one-dimensional or scalar source and target node embeddings, $s_i$ and $t_i$, which serve as the hub and authority score, respectively; we can express their relationship as $s_i = \sum_{i \mapsto j} t_j$ (i.e., good hubs point to good authorities) and $t_i = \sum_{j \mapsto i} s_j$ (i.e., good authorities are pointed to by good hubs).

Generalizing to our $d$-dimensional source and target encoding *vectors* $\mathbf{s}_i$ and $\mathbf{t}_i$, we could analogously write:

$$\mathbf{s}_i = \sum_j \mathbf{A}_{ij}\mathbf{t}_j \quad \text{and} \quad \mathbf{t}_i = \sum_j \mathbf{A}_{ji}\mathbf{s}_j. \tag{3}$$

We can write the above equations more compactly as $\mathbf{S} = \mathbf{AT}$ and $\mathbf{T} = \mathbf{A}^\top \mathbf{S}$. Conceptually, $\mathbf{s}_i$ and $\mathbf{t}_i$ play the role of multi-dimensional hub and authority scores.

**Implicit and Directed Adjacency via Attention:** The key insight in DiGT is that we should not rely on the fixed adjacency matrix $\mathbf{A}$; rather, we should construct an *implicit* adjacency matrix, denoted $\bar{\mathbf{A}}$, by exploiting the attention mechanism. A straightforward approach to compute $\bar{\mathbf{A}}$ could be $\bar{\mathbf{A}} = \mathbf{ST}^\top$. However, we need to make this learnable. To allow the flexibility of learning weight matrices for computing the implicit adjacency we use dual attention mechanisms. For the source nodes $\mathbf{S}$, let

$$\mathbf{Q}_S = \mathbf{S}\,\mathbf{W}_{QS} \qquad\qquad \mathbf{K}_S = \mathbf{S}\,\mathbf{W}_{KS} \qquad\qquad \mathbf{V}_S = \mathbf{S}\,\mathbf{W}_{VS} \tag{4}$$

and similarly for the target nodes $\mathbf{T}$, let

$$\mathbf{Q}_T = \mathbf{T}\,\mathbf{W}_{QT} \qquad\qquad \mathbf{K}_T = \mathbf{T}\,\mathbf{W}_{KT} \qquad\qquad \mathbf{V}_T = \mathbf{T}\,\mathbf{W}_{VT} \tag{5}$$

where all $\mathbf{W} \in \mathbb{R}^{d \times d_p}$ are learnable weight matrices, and $d_p$ is the projection dimensionality (suitably scaled down, based on the number of heads). We obtain a pair of attention matrices

$$\bar{\mathbf{A}}_{ST} = \left(\mathbf{Q}_S\mathbf{K}_T{}^\top\right)/\sqrt{d_p} \qquad\qquad \bar{\mathbf{A}}_{TS} = \left(\mathbf{Q}_T\mathbf{K}_S{}^\top\right)/\sqrt{d_p} \tag{6}$$

That is, the attention matrix $\bar{\mathbf{A}}_{ST}$ treats the source nodes as queries and the target as keys to compute their similarity, and vice-versa for $\bar{\mathbf{A}}_{TS}$.

**Edge Bias and Neighborhood Attention:** We now allow for the edge channels to directly influence the attention by introducing a per head *bias* matrix, $\mathbf{B}_{ST} \in \mathbb{R}^{n \times n}$, and *gate* matrix, $\mathbf{G}_{ST} \in \mathbb{R}^{n \times n}$, both of which are linear transformations from the edge encodings $\mathbf{E}_{ST}$ (with added layer norms). Further, $\mathbf{B}_{TS}$ and $\mathbf{G}_{TS}$ are their transpose matrices, respectively.

Next, we *localize* the attention from node channels to the $k$-hop neighborhood around each node. This is implemented by masking the attention matrix along with the edge bias via an element-wise product with the binary $k$-hop matrix $\mathbf{D}^{(k)}$ which is defined by setting $\mathbf{D}^{(k)}_{i,j} = 1$ *iff* $\delta_{ij} \leq k$ for the shortest path distance from node $i$ to node $j$, and zero otherwise. Thus, the attention matrices, denoted $\tilde{\mathbf{A}}$, for this layer are given as:

$$\tilde{\mathbf{A}}_{ST} = \left(\bar{\mathbf{A}}_{ST} + \mathbf{B}_{ST}\right) \odot \mathbf{D}^{(k)}_{ST} \qquad\qquad \tilde{\mathbf{A}}_{TS} = \left(\bar{\mathbf{A}}_{TS} + \mathbf{B}_{TS}\right) \odot \mathbf{D}^{(k)}_{TS} \tag{7}$$

This way the attention considers all node pairs within $k$-hops.

**Directional Attention:**   Finally, unlike traditional transformers that compute node importance via a softmax along each *row* of the attention matrix, we stack both $\tilde{\mathbf{A}}_{ST}$ and $\tilde{\mathbf{A}}_{TS}$ and compute the softmax along the stacking direction, given as

$$\tilde{\mathbf{A}}_{ST}, \tilde{\mathbf{A}}_{TS} = \texttt{softmax}(\tilde{\mathbf{A}}_{ST}, \tilde{\mathbf{A}}_{TS}). \tag{8}$$

By doing this, we weigh the importance of directionality for the attention/adjacency information.

Lastly, we enable the flow of information between nodes by gating their value representations prior to aggregation; this is realized as multiplication by the sigmoid function, $\sigma()$, of the entries in *gate* matrices, $\mathbf{G}_{ST}$ and $\mathbf{G}_{TS}$, resulting in

$$\mathbf{Y} = \left( (\tilde{\mathbf{A}}_{ST} \odot \sigma(\mathbf{G}_{ST})) \, \mathbf{V}_T \right) \, + \, \left( (\tilde{\mathbf{A}}_{TS} \odot \sigma(\mathbf{G}_{TS})) \, \mathbf{V}_S \right) \tag{9}$$

where $\mathbf{Y} \in \mathbb{R}^{n \times d_p}$ is the value representation for one head. So, when we have $h = d/d_p$ heads, we concatenate all of them (and add layer norm) to obtain the final value representation $\mathbf{Y} \in \mathbb{R}^{n \times d}$, for the next step. Also, the different DiGT layers do not share edge embeddings and this is also true for bias and gate matrices.

### 3.3   OUTPUT LAYERS AND PREDICTION

One point to note is that, after each DiGT layer, we take the combined value encoding $\mathbf{Y}$, and we use layer normalization and feed-forward network modules with residual connections, to produce the node and edge encoding outputs for a DiGT layer. These outputs become inputs for the next layer. Thus, the updated dual encodings $\mathbf{S}, \mathbf{T}$ for the next layer are given as:

$$\mathbf{S} = f(L_{VS}(\mathbf{Y})) \qquad\qquad \mathbf{T} = f(L_{VT}(\mathbf{Y})) \tag{10}$$

where, $L_{VS}$ and $L_{VT}$ are two linear transformations followed by a non-linear activation $f$ (with layer norms and residual connections). To obtain the updated edge embeddings $\mathbf{E}_{ST} \in \mathbb{R}^{n \times n \times d_e}$ for the next layer, we add together $\bar{\mathbf{A}}_{ST}$ and $\mathbf{B}_{ST}$ from all the $h$ heads, and apply a learnable linear transformation and non-linearity, as follows: $\mathbf{E}_{ST} = f(L_E(\bar{\mathbf{A}}_{ST} + \mathbf{B}_{ST}))$.

Lastly, to obtain the final output node embeddings, we concatenate both the source and target embeddings, as follows: $\mathbf{X} = \texttt{concat}(\mathbf{S}, \mathbf{T})$. After the last DiGT layer is processed, the encodings $\mathbf{X}$ are driven through some final learning task-specific modules. These are typically multilayer perceptron layers (MLP) for tasks related to *node* and *edge* learning (node classification, link prediction), or pooling layers for *graph-level* learning (graph classification, graph regression). For the directed graph classification task, we use *global average pooling* as our main method for producing a representation/encoding of the whole graph; this is essentially the average of the final node encodings. We also experiment with the method of *virtual nodes* based pooling (Hussain et al., 2022): a clique of artificial nodes (virtual nodes) are added to each graph and connected to all its nodes. We add bidirectional edges between each virtual node and all the rest of the graph nodes. After training, we average the concatenated source and target node embeddings of the virtual nodes and leverage the same final MLP layers for the downstream task.

### 3.4   ALTERNATIVE APPROACHES FOR DIRECTIONALITY

We now propose some other alternatives to model directionality directly in Transformer architectures.

**Exploiting Asymmetric Attention Matrix**   One simple approach to incorporating direction into the Transformer architecture is to leverage the inherent asymmetric nature of the attention matrix. Given node features $\mathbf{X} \in \mathbb{R}^{n \times d}$, the key, query, and value matrices are given as

$$\mathbf{Q} = \mathbf{X} \, \mathbf{W}_Q \qquad\qquad \mathbf{K} = \mathbf{X} \, \mathbf{W}_K \qquad\qquad \mathbf{V} = \mathbf{X} \, \mathbf{W}_V \tag{11}$$

Consider the attention matrix before the softmax, given as

$$\hat{\mathbf{A}} = (\mathbf{Q}\mathbf{K}^\top)/\sqrt{d_p}$$

This matrix is inherently asymmetric; we can thus exploit the attention that queries pay to keys and that keys pay to queries, to obtain the attention matrices by taking the softmax along the rows (same as regular attention) or softmax along the columns, as follows

$$\bar{\mathbf{A}}_{ST} = \texttt{softmax}(\hat{\mathbf{A}}) \qquad \bar{\mathbf{A}}_{TS} = \texttt{softmax}_{\texttt{cols}}(\hat{\mathbf{A}}) = \texttt{softmax}(\hat{\mathbf{A}}^\top) \qquad (12)$$

where we use the subscript `cols` to denote that softmax is applied along the columns.

To obtain the new value matrix several different strategies can be used to aggregate the two attention matrices together, which we discuss in the ablation studies. An effective approach is to do a weighted (learnable) sum of the two value representations:

$$\mathbf{Y} = \mathbf{W}_{ST}\, \bar{\mathbf{A}}_{ST}\, \mathbf{V} + \mathbf{W}_{TS}\, \bar{\mathbf{A}}_{TS}\, \mathbf{V} \qquad (13)$$

where $\mathbf{W}_{ST}, \mathbf{W}_{TS} \in \mathbb{R}^{d \times d}$ are learnable weight matrices. Note that we can also leverage directional attention (see Eq. (8)) in the equation above.

**Exploiting Dual Attention Matrices** Another strategy to incorporate direction is by leveraging two key, query and value representations for each node, given as:

$$\mathbf{Q}_S = \mathbf{X}\,\mathbf{W}_{QS} \qquad \mathbf{K}_S = \mathbf{X}\,\mathbf{W}_{KS} \qquad \mathbf{V}_S = \mathbf{X}\,\mathbf{W}_{VS} \qquad (14)$$
$$\mathbf{Q}_T = \mathbf{X}\,\mathbf{W}_{QT} \qquad \mathbf{K}_T = \mathbf{X}\,\mathbf{W}_{KT} \qquad \mathbf{V}_T = \mathbf{X}\,\mathbf{W}_{VT} \qquad (15)$$

We can then obtain dual attention matrices:

$$\bar{\mathbf{A}}_{ST} = \texttt{softmax}\left((\mathbf{Q}_S \mathbf{K}_T{}^\top)/\sqrt{d_p}\right) \qquad \bar{\mathbf{A}}_{TS} = \texttt{softmax}((\mathbf{Q}_T \mathbf{K}_S{}^\top)/\sqrt{d_p}) \qquad (16)$$

which can then be aggregated to obtain the new value representation via the weighted sum approach in Eq. (13).

This is similar to DiGT, but the key difference is that the dual representations are used only within the attention layer, whereas DiGT uses dual node embeddings in all layers.

## 4 EXPERIMENTS

We conduct our experiments mainly on NVIDIA V100 GPUs, with 32GB memory, using PyTorch. An anonymous link to our implementation is provided in the Appendix, which also contains additional experimental details and ablation studies.

### 4.1 DIRECTED GRAPH DATASETS

There are several well-known directed datasets to assess the performance of graph models, such as `MNIST` (Achanta et al., 2012)and `CIFAR10` (Krizhevsky et al., 2009), `Ogbg-Code2` (Hu et al., 2020), and `Malnet-tiny` (Freitas et al., 2020). To evaluate the directionality in these datasets, we design a random flip test. In essence, for a given edge flip probability, say $\theta$ (e.g., $\theta \in \{0.25, 0.5\}$), given edge $(u, v)$ we flip it with probability $\theta$ during each training, validation and testing step. If a model consistently achieves accuracy comparable to the original dataset, it suggests that directionality is not a crucial factor. Table 1 shows the results with $\theta = 0.5$ (i.e, 50% of the edges are flipped); see Appendix for full results. We use EGT (Hussain et al., 2022) on the `MNIST` and `CIFAR10` graphs, Exphormer (Shirzad et al., 2023), which is the SOTA on `Malnet-tiny`, and DAGformer (Luo, 2022), which is the top-performer on the `Ogbg-Code2` leaderboard (Hu et al., 2020). We observe that despite altering the direction of edges on `MNIST`, `CIFAR10` and `Ogbg-Code2`, the results remain largely unaffected, which indicates that directionality is not important. On the other hand, there is a performance loss on `Malnet-tiny`.

**FlowGraph datasets.** Given the limitations of existing benchmarks, we introduce a family of directed graph datasets that explicitly relate the edge direction pattern in graphs to their classification labels. In particular, we generate graphs with their nodes organized in successive layers and then we leverage the notion of a flow between the layers through directed edges: for a predefined subset of layers, graphs with different aggregate flow between successive layers in the subset are assigned different labels. Our `FlowGraph` generator is modeled after the *Directed Stochastic Block Model*

Table 1: Randomized directionality via edge flips: Model performance

| Model | MNIST | CIFAR10 | Model | Ogbg-Code2 | Model | Malnet-tiny |
|---|---|---|---|---|---|---|
| EGT | 98.41 +/- 0.04 | 68.70 +/- 0.41 | DAG | 20.2 +/- 0.2 | Exphormer | 94.02 +/- 0.21 |
| EGT-Flip50 | 97.99 +/- 0.09 | 67.28 +/- 0.38 | DAG-Flip50 | 19.0 +/- 0.1 | Exphormer-Flip50 | 87.90 +/- 1.65 |

| Model | FlowGraph2 | FlowGraph3 | FlowGraph6 | Twitter3 | Twitter5 |
|---|---|---|---|---|---|
| DiGT | 97.42 +/- 0.82 | 74.55 +/- 0.69 | 46.80 +/- 0.97 | 91.67 +/- 0.79 | 85.94 +/- 0.25 |
| DiGT-Flip50 | 49.67 +/- 1.39 | 32.33 +/- 1.66 | 16.78 +/- 0.04 | 82.96 +/- 1.13 | 65.44 +/- 0.38 |

*(DSBM)* (Malliaros & Vazirgiannis, 2013). Following the notation in (He et al., 2021), we organize $N$ graph nodes into $K$ clusters and define cluster adjacencies in a meta-graph adjacency matrix $\mathbf{F}$, with its entries $\mathbf{F}_{kl}$ marking the allowance of directed edges from nodes in cluster $k$ to those of cluster $l$. More specifically, we assume that the node clusters are arranged sequentially, $l = 0, 1, \ldots, K-1$ (say from left to right) and a subset of its first $l_S < K$ consecutive clusters define a subgraph $S$. In `FlowGraph` we allow directed edges between nodes belonging to all clusters with the probability being a small noise parameter $\eta$ (typically $\eta = 0.01$). Then for directed edges between nodes in successive clusters, with the source node $l$ being in a cluster in subgraph $S$, we set $F_{l,l+1}$ to a percentage $f\%$. These percentages are different for different classes and depend on the number of classes $n_c$. In our experiments, for all generated graphs we set $N = 150$, $K = 10$, $l_S = 4$. We generate 3 graph datasets: one dataset for each of the $n_c = 2, 3, 6$-class cases. We depict three graph instances from each of the 3 classes of `FlowGraph3` in the Appendix.

**Twitter datasets.** We use 973 directed ego-networks from Twitter[1], each corresponding to some user $u$ (*ego*): the ego-network is between $u$'s friends also referred to as *alters* (Leskovec & Mcauley, 2012). If nodes $v_i$, $v_j$ are in $u$'s ego-network then $u$ follows them and if $v_i$ follows $v_j$ then there is a directed edge $v_i \mapsto v_j$ in the ego-network. We introduce perturbations to each of these real ego-networks where a perturbation can be either (i) *rewiring* of an existing edge (an $(a, b) \in E(ego(u))$, where it is deleted and replaced by an edge $(c, d)$ where nodes $c, d$ are randomly selected from $V(ego(u))$), or (ii) *reversing* of the direction of an existing edge $(a, b) \in E(ego(u))$, where it is replaced by $(b, a)$. The percentage of the perturbed edges in an ego-network can be $[0, 25, 50, 75, 100]\%$. Rewiring and reversing the direction of edges takes place with equal probabilities. So, for each of the percentages, 973 new perturbed ego-networks are generated, each labeled with the corresponding perturbation percentage. We refer to the collection of the $5 \times 973$ perturbed `Twitter` datasets as `Twitter5` (5 labels/classes). Similarly, if we get $3 \times 973$ of them corresponding to perturbation percentages $[0\%, 50\%, 100\%]$, then we have the `Twitter3` dataset (3 labels/classes).

As we can see from Table 1, there is a significant drop in performance when we randomly flip the edges for both `Flowgraph` and `Twitter` datasets, thus direction is important.

**Degree of Directionality** To characterize the directionality of a dataset, we defined another measure, called the *degree of directionality* for a graph. Let SCC denote a strongly connected component in a directed graph (a maximal subset of mutually reachable nodes). Further, given the set of $m$ SCCs of a directed graph, $\mathcal{S} = \{S_1, S_2, ..., S_m\}$, define the *SCC entropy* of the graph as follows: $E(\mathcal{S}) = -\sum_{i=1}^{m} p_i \log p_i$, where $p_i = |S_i|/n$. A low entropy means that most nodes are mutually reachable, and thus directionality is not expected to play a big role. On the other hand, if the SCC entropy is $\log n$, like for a directed acyclic graph, then directionality is clearly important.

Figure 1 plots the SCC entropy for the different benchmark datasets; for each class, we plot the average and standard deviation. We can see a very clear trend. `FlowGraph` classes are inherently directed, with larger entropies, and `Twitter` captures the inherent directionality of the "follow" relationship between entities, exhibiting lower entropy values. As we increase perturbation levels, entropy decreases as expected due to the random modification of the original directed graph structures. More importantly, the derived `MNIST` and `CIFAR10` have no inherent directedness, and their

---

[1] https://snap.stanford.edu/data/ego-Twitter.html

entropy values are extremely low. These results justify our choice to restrict the directed graphs benchmarks to only those where direction matters. Furthermore, we will show that DiGT performs even better when direction matters more.

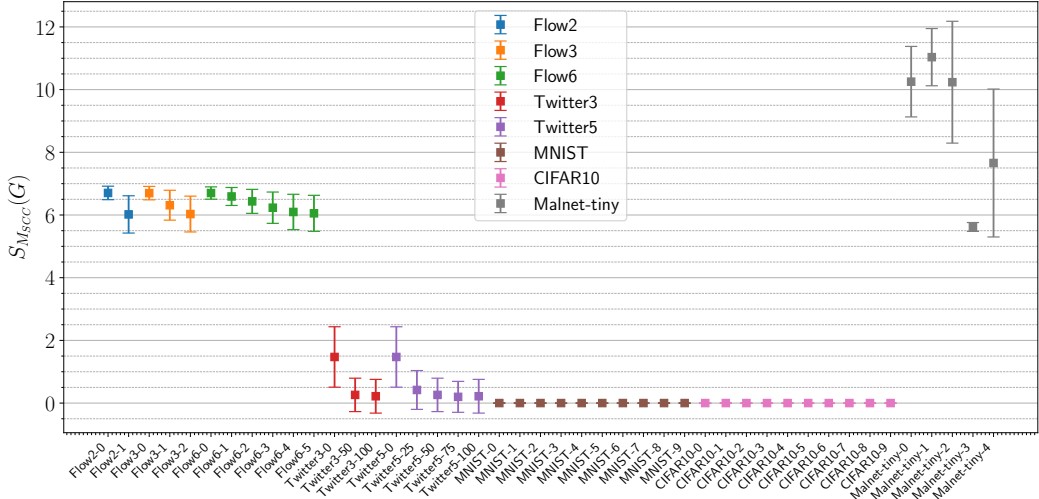

Figure 1: SCC Entropy Plot: The entropy for each class (by increasing perturbation or class label) for each dataset is shown. For `Flowgraph` the perturbation refers to % of right to left edges, for `Twitter` its random edge rewirings.

Table 2: Accuracy for GNN and GT models. Blank denotes no previous reported results.

| Model | FlowGraph2 | FlowGraph3 | FlowGraph6 | Twitter3 | Twitter5 | Malnet-tiny |
|---|---|---|---|---|---|---|
| GCN (Kipf & Welling, 2016) | 87.50 +/- 1.27 | 58.28 +/- 0.88 | 30.36 +/- 0.55 | 76.24 +/- 0.56 | 61.23 +/- 1.67 | |
| GAT (Veličković et al., 2017) | 84.92 +/- 1.90 | 58.83 +/- 1.47 | 30.31 +/- 0.28 | 74.59 +/- 1.59 | 56.79 +/- 0.05 | 92.1 +/- 0.24 |
| DiGCN (Tong et al., 2020a) | 95.67 +/- 0.51 | 71.22 +/- 1.03 | 36.78 +/- 0.85 | 73.51 +/- 0.61 | 52.85 +/- 1.94 | |
| PNA (Corso et al., 2020) | 96.17 +/- 0.31 | 72.94 +/- 0.64 | 41.42 +/- 1.32 | 88.26 +/- 1.16 | 70.94 +/- 2.01 | |
| Graph Transformer (Dwivedi & Bresson, 2021) | 93.17 +/- 0.82 | 66.17 +/- 0.60 | 36.20 +/- 1.12 | 90.66 +/- 0.35 | 79.55 +/- 0.68 | |
| SAN (Kreuzer et al., 2021) | 91.73 +/- 1.84 | 63.87 +/- 0.66 | 34.57 +/- 0.45 | 85.33 +/- 0.78 | 63.13 +/- 1.65 | |
| EGT (Hussain et al., 2022) | 95.00 +/- 1.67 | 72.06 +/- 1.16 | 42.87 +/- 0.62 | 86.49 +/- 0.73 | 73.94 +/- 1.47 | |
| Exphormer (Shirzad et al., 2023) | 96.72 +/- 0.44 | 72.81 +/- 0.38 | 41.70 +/- 0.39 | 89.76 +/- 0.30 | 72.72 +/- 1.40 | 94.02 +/- 0.21 |
| DiGT | **97.42 +/- 0.82** | **74.55 +/- 0.69** | **46.80 +/- 0.97** | **91.67 +/- 0.79** | **85.94 +/- 0.25** | |

## 4.2 EXPERIMENTAL COMPARISON

### 4.2.1 DiGT vs. GNNs AND GTs

We now show experimental results on the directed graph datasets where directionality is important. We select GCN (Kipf & Welling, 2016), GAT (Veličković et al., 2017), DiGCN (Tong et al., 2020a), and PNA (Corso et al., 2020), for our comparisons with graph neural networks, and Graph Transformer (Dwivedi & Bresson, 2021), SAN (Kreuzer et al., 2021), EGT (Hussain et al., 2022), and Exphormer (Shirzad et al., 2023) for our comparisons with graph transformers. We take the results GAT and Exphormer with the `Malnet-tiny` dataset from (Shirzad et al., 2023). We choose EGT (Hussain et al., 2022) as the representative of graph transformers with global dense attention, and Exphormer (Shirzad et al., 2023) as the representative of graph transformers with local sparse attention. We compare these approaches with our DiGT approach.

The accuracy results are listed in Table 2. The methods are grouped by GNNs, GTs, and finally DiGT. We can see that our DiGT model clearly outperforms all other GNN and GT models, by significant margins, especially as the directionality becomes more important, i.e., DiGT has better margins for `FlowGraph6`, `Twitter3` and `Twitter5`. DiGT employs global dense attention (like EGT), making it adept at capturing more complex graph structures, but the explicit directional attention makes it shine on all the datasets. On the contrary, Exphormer utilizes local attention

by leveraging message passing to nearby neighbors. Like some of the GNNs, one advantage of Exphormer is that its use of expander graphs allows it to run on the larger `Malnet-tiny` dataset (which has some graphs over 2000 nodes), where other GT methods, including DiGT cannot be applied (incorporating sparse graph connectivity into DiGT is part of our future work). As such, these results clearly demonstrate the effectiveness of the directional strategies employed in DiGT, especially the inherent dual source and target encodings, with directional attention and restricted $k$-hop neighborhood.

Table 3: Incorporating directionality into graph transformers. Bold denotes best results, italics the second best.

| Model | FlowGraph2 | FlowGraph3 | FlowGraph6 | Twitter3 | Twitter5 | Malnet-tiny |
|---|---|---|---|---|---|---|
| Vanilla Transformer | 95.58 +/- 0.66 | 68.22 +/- 0.55 | 39.56 +/- 1.61 | 89.12 +/- 0.43 | 77.03 +/- 1.40 | |
| Vanilla-asym | 95.50 +/- 0.20 | 69.28 +/- 0.75 | 41.42 +/- 0.78 | *91.28 +/- 0.78* | 79.42 +/- 0.48 | |
| EGT (Hussain et al., 2022) | 95.00 +/- 1.67 | 72.06 +/- 1.16 | 42.87 +/- 0.62 | 86.49 +/- 0.73 | 73.94 +/- 1.47 | |
| EGT-asym | 96.50 +/- 0.89 | 72.39 +/- 0.55 | *42.88 +/- 0.20* | 90.37 +/- 0.29 | *82.60 +/- 0.35* | |
| Exphormer (Shirzad et al., 2023) | 96.72 +/- 0.44 | 72.81 +/- 0.38 | 41.70 +/- 0.39 | 89.76 +/- 0.30 | 72.72 +/- 1.40 | 94.02 +/- 0.21 |
| Exphormer-asym | **98.41 +/- 1.18** | **74.72 +/- 2.24** | 42.67 +/- 0.80 | 90.83 +/- 1.14 | 80.07 +/- 1.18 | 93.85 +/- 0.15 |
| Exphormer-dual | 97.42 +/- 0.31 | 73.39 +/- 0.90 | 42.28 +/- 0.17 | 90.78 +/- 0.81 | 79.59 +/- 0.69 | **94.23 +/- 0.20** |
| DiGT | *97.42 +/- 0.82* | *74.55 +/- 0.69* | **46.80 +/- 0.97** | **91.67 +/- 0.79** | **85.94 +/- 0.25** | |

### 4.2.2 Incorporating Direction into GTs

To show the effectiveness of directionality-based strategies outlined in Section 3, we modify the vanilla Transformer (Vaswani et al., 2017), EGT, and Exphormer models and convert these "undirected" graph transformer models into directed ones. Using the asymmetric attention matrix (Eq. (12)) is denoted with the suffix '-asym', and using the dual attention matrices (Eq. (16)) is denoted by suffix '-dual'. Table 3 shows the baseline model, and the modification using the best performing directionality approach.

Interestingly, the directed version *always* outperforms its corresponding baseline graph transformer model. This shows the power of incorporating direction on those datasets where it really matters. In fact, the Exphormer-dual outperforms the baseline Exphormer model, and results in new SOTA result. When we compate DiGT with these models, we again find that DiGT outperforms all models (see bold results) on the datasets with more directionality, like `FlowGraph6`, `Twitter3` and `Twitter5`. Even on `FlowGraph2` and `FlowGraph3`, DiGT is the second best model, only slightly behind Exphormer-asym, which is also one of our own direction-based strategies.

## 5 Conclusions and Future Work

In this paper, we present DiGT, a novel architecture for capturing graph directionality using transformers. We empirically evaluate its classification accuracy on directional graph datasets and demonstrate its superior performance against state-of-the-art Graph Transformers and Graph Neural Networks. We also propose other strategies that can be used to add "direction" to other graph transformer models. Doing so yields a new SOTA result on the `Malnet-tiny` dataset. In fact, our directed extensions always outpeform their undirected baselines. To our knowledge, we are the first to point out the limitations of some of the directed graph benchmarks, in that direction does not seem to matter for the classification task. We therefore propose new directed benchmark datasets, and show the superior performace on DiGT on those graphs.

One limitation of our DiGT model is that due to the quadratic complexity of attention, like most Transformers, it does not scale to larger graphs. To scale up the attention mechanism, we plan to explore techniques like expander graphs used in Exphormer (Shirzad et al., 2023), and other approaches to expand the context (Bertsch et al., 2023; Tay et al., 2022), and study their effectiveness for (directed) graph datasets.

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
