# DIRECTIONALITY IN GRAPH TRANSFORMERS

## A EXPERIMENT DETAILS

We test our code on a node with NVIDIA V100 GPUs (32GB RAM), 20-core 2.5Ghz Intel Xeon CPU (768GB RAM), running Linux. We use Python and specifically the PyTorch library for our implementation. Our code is available via the anonymous Google Drive Link: `https://drive.google.com/file/d/193Tf8Mk6cen8kEwKfzP-zmSnkQDJzQBw/view?usp=sharing`

For fair comparison, we keep all the experiments with the number of learnable parameters around $100K$ with 4 layers for the graph classification tasks. We fix the batch size to 32, the number of maximum epochs to 200, and we employ grid search for tuning the learning rate $\eta \in \{2^i \times u | i = 0, 1, 2, 3, 4\}$ with $u = 5 \times 10^{-4}$, and $\eta = 8 \times 10^{-3}$ (or $i = 4$). We employ grid search for tuning the number of hops $k \in \{i | 1 \le i \le 9\}$. We apply a similar process to other models.

We use GCN (Kipf & Welling, 2016) and GAT (Veličković et al., 2017) from the Benchmarking Graph Neural Networks framework (Dwivedi et al., 2020); we use DGCN (Tong et al., 2020b), DiGCN (Tong et al., 2020a), and DiGCNIB (Tong et al., 2020a) from PyTorch Geometric Signed Directed framework (He et al., 2022), and we run PNA (Corso et al., 2020), Graph Transformer (Dwivedi & Bresson, 2021), SAN (Kreuzer et al., 2021), and EGT (Hussain et al., 2022) using their provided implementations. Table 1 contains parameters used during experimentation.

For the MNIST (Achanta et al., 2012), CIFAR10 (Krizhevsky et al., 2009), Ogbg-Code2 (Hu et al., 2020), and Malnet-tiny (Freitas et al., 2020) datasets, we follow the exact experimental settings from the provided papers and codes to ensure a fair comparison.

Table 1: Training Settings (default values)

| Hyperparameters | FlowGraph Twitter |
|---|---|
| Batch Size | 32 |
| Number of Epochs | 200 |
| Early Stops | 0 |
| Max Learning Rate ($\eta$) | 0.008 |
| Number of Virtual Nodes ($q$) | 0 |
| Number of Layers | 4 |
| Number of Heads ($h = d/d_p$) | 8 |
| Node dimensionality ($d$) | 32 |
| Edge dimensionality ($d_e$) | 32 |
| Projection dimensionality ($d_p$) | 4 |
| SVD dimension ($r$) | 8 |

## B DATASET DETAILS

Table 2 shows the dataset statistics for each of the datasets used in our experiments. These include the number of graph instances, average number of nodes and edges, and the number of graph classes.

Table 2: Datasets details: Number of graph instances $N$ in the dataset, average number of nodes $n$, average number of directed edges $e$ and the number of graph classes $n_c$ are tabulated

| Dataset | $N$ | Avg. $n$ | Avg. $e$ | $n_c$ |
|---|---|---|---|---|
| FlowGraph2 | 2000 | 114.46 | 150.30 | 2 |
| FlowGraph3 | 3000 | 111.66 | 150.70 | 3 |
| FlowGraph6 | 6000 | 111.40 | 149.93 | 6 |
| Twitter3 | 2919 | 131.76 | 2237.55 | 3 |
| Twitter5 | 4865 | 131.76 | 2208.79 | 5 |
| Malnet-tiny | 5000 | 1410.3 | 2859.9 | 6 |
| MNIST | 70000 | 70.57 | 564.53 | 10 |
| CIFAR10 | 60000 | 117.63 | 941.07 | 10 |
| Ogbg-Code2 | 452,741 | 125.2 | 124.2 | 5 subtokens |

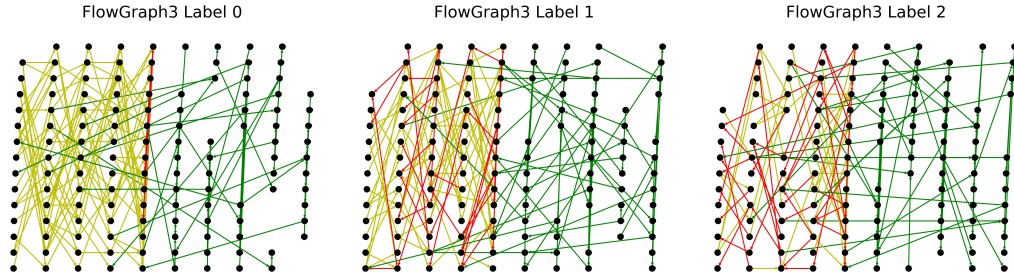

Figure 1: The visualization of three categories of graphs in `FlowGraph3` dataset.

## B.1 FLOWGRAPH VISUALIZATION

Figure 1 visualizes FlowGraph 3. The yellow edges are the flows from left to right, the red edges are the flows from right to left, and the green edges are noises. In all three samples, half of the edges are noises with green edges. In the left sample, besides the noise edges, almost all the edges have the flow from left to right; in the middle sample, $75\%$ of the edges have the flow from left to right; whereas in the right sample, $50\%$ of the edges have the flow from left to right.

## C  MEASURING IMPORTANCE OF DIRECTIONALITY

**Flipping Edge Directions**  We reverse the direction of $25\%$ and $50\%$ of graph edges, randomly selected, and empirically evaluate the importance of directionality in all the datasets. Table 3 list our findings. We use GAT(Veličković et al., 2017) and EGT (Hussain et al., 2022) as the representatives for graph neural networks and graph transformers. We confirm that the *derived* notion of *edge direction* in `MNIST` (Achanta et al., 2012) and `CIFAR10` (Krizhevsky et al., 2009) is not significant: classification results from both EGT and GAT models are almost agnostic to edge direction flips in these datasets; the differences between the vanilla datasets and flipped datasets are at most $2.1\%$.

For the `Ogbg-Code2` dataset (Hu et al., 2020), we select SAT (Chen et al., 2022) and DAGformer (Luo, 2022), two top-performing models, as the baseline for testing random flips. We observe that DAG only exhibits a $1.2\%$ decrease in performance when $50\%$ of the edges in this dataset are randomly flipped. This suggests that directionality is not a significant factor in this dataset.

We employ the current state-of-the-art model, Exphormer(Shirzad et al., 2023), to examine the significance of direction in `Malnet-tiny` dataset (Freitas et al., 2020). It is observed that there's a $6.12\%$ decrease in performance in the flip50 case. This gap refers to the importance of directionality in the `Malnet-tiny` dataset.

For `FlowGraph` and `Twitter` datasets, the direction of edges is enforced by construction or emerges naturally and is thus expected to correlate strongly with graph labels. In particular, DiGT accuracy *consistently* decreases across all `FlowGraph` and `Twitter` datasets and all edge reversal percentages. In `FlowGraph`, accuracy drops are sharper (and saturate as we increase the reversal percentage): with 25% of edges flipped accuracy decreases in the range of 46.25% to 28.94%: from 97.42% to 51.17% in `FlowGraph2` and from 46.80% to 17.86% in `FlowGraph6`.

Table 3: *Random Flips of* `MNIST`, `CIFAR10`, `Ogbg-Code2`, `Malnet-tiny` *datasets.*

| Model | MNIST | CIFAR10 | Model | Ogbg-Code2 | Model | Malnet-tiny |
|---|---|---|---|---|---|---|
| GAT | 95.54 +/- 0.21 | 64.22 +/- 0.46 | SAT | 19.37 +/- 0.03 | Exphormer | 94.02 +/- 0.21 |
| GAT-Flip25 | 93.92 +/- 0.22 | 62.86 +/- 0.18 | SAT-Flip25 | 18.72 +/- 0.08 | Exphormer-Flip25 | 88.77 +/- 0.41 |
| GAT-Flip50 | 93.43 +/- 0.23 | 62.11 +/- 0.78 | SAT-Flip50 | 18.70 +/- 0.03 | Exphormer-Flip50 | 87.90 +/- 1.65 |
| EGT | 98.41 +/- 0.04 | 68.70 +/- 0.41 | DAG | 20.2 +/- 0.2 | | |
| EGT-Flip25 | 97.90 +/- 0.11 | 67.27 +/- 0.56 | DAG-Flip25 | 18.9 +/- 0.2 | | |
| EGT-Flip50 | 97.99 +/- 0.09 | 67.28 +/- 0.38 | DAG-Flip50 | 19.0 +/- 0.1 | | |

Table 4: *Random Flips of* `FlowGraph` *and* `Twitter` *datasets.*

| Model | FlowGraph2 | FlowGraph3 | FlowGraph6 | Twitter3 | Twitter5 |
|---|---|---|---|---|---|
| GAT | 84.92 +/- 1.90 | 58.83 +/- 1.47 | 30.31 +/- 0.28 | 74.59 +/- 1.59 | 56.79 +/- 0.05 |
| GAT-Flip25 | 73.67 +/- 0.72 | 44.16 +/- 1.70 | 48.62 +/- 0.80 | 67.69 +/- 1.06 | 48.62 +/- 0.80 |
| GAT-Flip50 | 51.00 +/- 3.89 | 33.00 +/- 2.38 | 17.31 +/- 0.66 | 65.64 +/- 1.71 | 44.34 +/- 2.05 |
| DiGT | 97.42 +/- 0.82 | 74.55 +/- 0.69 | 46.80 +/- 0.97 | 91.67 +/- 0.79 | 85.94 +/- 0.25 |
| DiGT-Flip25 | 51.17 +/- 1.25 | 33.50 +/- 0.36 | 17.86 +/- 0.92 | 89.29 +/- 0.90 | 77.47 +/- 0.90 |
| DiGT-Flip50 | 49.67 +/- 1.39 | 32.33 +/- 1.66 | 16.78 +/- 0.04 | 82.96 +/- 1.13 | 65.44 +/- 0.38 |

# D    DETAILED COMPARISON OF DIRECTIONAL TRANSFORMER MODELS

We study how direction can be incorporated in the undirected graph Transformer architectures using the approaches outlined in **??**. In particular, we try three variants. The first is the use of the asymmetric attention matrix, denoted by the suffix '-asym', and the second is the use of dual encodings, denoted by '-dual' from **??**. The third approach is to use the directional attention that is also used in DiGT, denoted by the suffix '-DA', i.e., taking softmax across the direction axis.

We compare the variants on `FlowGraph` and `Twitter` datasets on the Vanilla Transformer (Vaswani et al., 2017) and EGT (Hussain et al., 2022) models, and also use `Malnet-tiny` on the Exphormer (Shirzad et al., 2023) model.

From Table 5, we can observe that Vanilla-asym has the best performance on most of the datasets, with second best on `FlowGraph2`. Table 6 shows that EGT-asym is once again the most effective variant. Finally, when we incorporate the different directionality approaches into Exphormer, once again the asymmetric attention approach yields the best results on the `FlowGraph` and `Twitter` datasets. Interestingly, as seen in Table 7, Exphormer is the only method that can run on larger graphs like in `Malnet-tiny`, where Exphormer-dual results in suprior performance. In fact, since it beats the baseline Exphormer, this represents a new SOTA result on this dataset.

# E    ABLATION STUDY DETAILS

## E.1    DIGT VARIANTS

We also explored what happens to the performance of DiGT if we use dual encoding approach, or if we do not use the $k$-hop neighborhood. Since DiGT is inherently a dual encoding based

Table 5: Classification accuracy of Vanilla Graph Transformer and its variants.

| Model | FlowGraph2 | FlowGraph3 | FlowGraph6 | Twitter3 | Twitter5 |
|---|---|---|---|---|---|
| Vanilla Transformer | **95.58 +/- 0.66** | 68.22 +/- 0.55 | 39.56 +/- 1.61 | 89.12 +/- 0.43 | 77.03 +/- 1.40 |
| Vanilla-sym | 95.50 +/- 0.20 | **69.28 +/- 0.75** | **41.42 +/- 0.78** | **91.28 +/- 0.78** | **79.42 +/- 0.48** |
| Vanilla-dual | 94.92 +/- 0.51 | 65.72 +/- 0.55 | 38.36 +/- 1.34 | 90.43 +/- 0.48 | 79.93 +/- 0.21 |
| Vanilla-DA | 93.58 +/- 1.01 | 68.22 +/- 1.23 | 38.08 +/- 0.82 | 66.10 +/- 2.53 | 44.85 +/- 3.40 |

Table 6: Classification accuracy of EGT and its variants

| Model | FlowGraph2 | FlowGraph3 | FlowGraph6 | Twitter3 | Twitter5 |
|---|---|---|---|---|---|
| EGT (Hussain et al., 2022) | 95.00 +/- 1.67 | 72.06 +/- 1.16 | 42.87 +/- 0.62 | 86.49 +/- 0.73 | 73.94 +/- 1.47 |
| EGT-asym | **96.50 +/- 0.89** | **72.39 +/- 0.55** | **42.88 +/- 0.20** | **90.37 +/- 0.29** | **82.60 +/- 0.35** |
| EGT-dual | 95.17 +/- 1.33 | 71.83 +/- 2.38 | 42.81 +/- 1.02 | 90.14 +/- 0.43 | 79.66 +/- 0.79 |
| EGT-DA | 96.48 +/- 0.42 | 70.83 +/- 2.68 | 42.92 +/- 0.59 | 87.07 +/- 1.91 | 77.88 +/- 2.06 |

Table 7: Classification accuracy of Exphormer and its variants

| Model | FlowGraph2 | FlowGraph3 | FlowGraph6 | Twitter3 | Twitter5 | Malnet-tiny |
|---|---|---|---|---|---|---|
| Exphormer | 96.72 +/- 0.44 | 72.81 +/- 0.38 | 41.70 +/- 0.39 | 89.76 +/- 0.30 | 72.72 +/- 1.40 | 94.02 +/- 0.21 |
| Exphormer-sym | **98.41 +/- 1.18** | **74.72 +/- 2.24** | **42.67 +/- 0.80** | **90.83 +/- 1.14** | **80.07 +/- 1.18** | 93.85 +/- 0.15 |
| Exphormer-dual | 97.42 +/- 0.31 | 73.39 +/- 0.90 | 42.28 +/- 0.17 | 90.78 +/- 0.81 | 79.59 +/- 0.69 | **94.23 +/- 0.20** |
| Exphormer-DA | 97.92 +/- 0.82 | 74.67 +/- 0.83 | 42.45 +/- 0.12 | 90.54 +/- 0.49 | 79.83 +/- 1.35 | 93.60 +/- 0.35 |

method, we did not try the asymmetric single attention matrix approach. As such, DiGT-dual uses dual key, query and value matrices only during the attention computation, whereas DiGT has dual embeddings in all the layers including the input, feed-forward and output layers. DiGT-DA (no $k$-hops) still uses directional attention, but without the $k$-hop mechanism. Table 8 shows that DiGT-DA (no $k$-hops) surpasses DiGT-asym across all datasets. Nevertheless, restricting the attention to $k$-hop neighborhood, the default DiGT strategy, yields the best overall results on all the datasets.

Table 8: Classification accuracy of DiGT and its variants.

| Model | FlowGraph2 | FlowGraph3 | FlowGraph6 | Twitter3 | Twitter5 |
|---|---|---|---|---|---|
| EGT (Hussain et al., 2022) | 95.00 +/- 1.67 | 72.06 +/- 1.16 | 42.87 +/- 0.62 | 86.49 +/- 0.73 | 73.94 +/- 1.47 |
| DiGT-dual | 95.42 +/- 1.36 | 72.17 +/- 1.30 | 44.06 +/- 0.57 | 87.86 +/- 2.06 | 81.13 +/- 0.15 |
| DiGT-DA (no $k$-hops) | 96.52 +/- 1.39 | 74.33 +/- 0.54 | 46.22 +/- 1.74 | 91.31 +/- 1.14 | 85.46+/- 1.26 |
| DiGT | **97.42 +/- 0.82** | **74.55 +/- 0.69** | **46.80 +/- 0.97** | **91.67 +/- 0.79** | **85.94 +/- 0.25** |

## E.2 ABLATION STUDY: DOUBLE SOFTMAX

We were also curious about whether implementing two softmax functions — a standard softmax to capture neighbor relations, and a stacking softmax to capture directionality — could enhance the overall performance. The algorithm is shown at the Equation 1 Therefore, we conduct experiments using the vanilla transformer, EGT, and Exphormer with dual softmax configurations. However, as Table 9 demonstrates, utilizing two softmax functions does not aid the models in capturing the directionalities within graphs. The interaction between the two softmax functions appear to adversely

affect each other, leading to worse performance.

$$\bar{\mathbf{A}}_{ST} = \mathtt{softmax}\left((\mathbf{Q}_S\mathbf{K}_T{}^\top)/\sqrt{d_p}\right) \qquad \bar{\mathbf{A}}_{TS} = \mathtt{softmax}((\mathbf{Q}_T\mathbf{K}_S{}^\top)/\sqrt{d_p}) \qquad (1)$$

$$\bar{\mathbf{A}}_{ST}, \bar{\mathbf{A}}_{TS} = \mathtt{softmax}(\bar{\mathbf{A}}_{ST}, \bar{\mathbf{A}}_{TS}).$$

$$\mathbf{Y} = \mathbf{W}_{ST}\,\bar{\mathbf{A}}_{ST}\,\mathbf{V} + \mathbf{W}_{TS}\,\bar{\mathbf{A}}_{TS}\,\mathbf{V}$$

where $\mathbf{W}_{ST}, \mathbf{W}_{TS} \in \mathbb{R}^{d \times d}$ are learnable weight matrices.

Table 9: Classification accuracy using Double Softmax

| Model | FlowGraph2 | FlowGraph3 | FlowGraph6 | Twitter3 | Twitter5 | Malnet-tiny |
|---|---|---|---|---|---|---|
| Vanilla Transformer | **95.58 +/- 0.66** | 68.22 +/- 0.55 | 39.56 +/- 1.61 | 89.12 +/- 0.43 | 77.03 +/- 1.40 | |
| Vanilla-asym | 95.50 +/- 0.20 | **69.28 +/- 0.75** | **41.42 +/- 0.78** | **91.28 +/- 0.78** | **79.42 +/- 0.48** | |
| Vanilla-2softmax | 50.83 +/- 1.70 | 33.72 +/- 2.84 | 16.69 +/- 0.58 | 66.38 +/- 5.96 | 47.21 +/- 2.96 | |
| EGT | 95.00 +/- 1.67 | 72.06 +/- 1.16 | 42.87 +/- 0.62 | 86.49 +/- 0.73 | 73.94 +/- 1.47 | |
| EGT-asym | **96.50 +/- 0.89** | 72.39 +/- 0.55 | **42.88 +/- 0.20** | **90.37 +/- 0.29** | **82.60 +/- 0.35** | |
| EGT-2softmax | 95.08 +/- 1.30 | **72.50 +/- 0.76** | 42.44 +/- 0.79 | 89.57 +/- 1.14 | 81.13 +/- 1.78 | |
| Exphormer | 96.72 +/- 0.44 | 72.81 +/- 0.38 | 41.70 +/- 0.39 | 89.76 +/- 0.30 | 72.72 +/- 1.40 | 94.02 +/- 0.21 |
| Exphormer-asym | **98.41 +/- 1.18** | **74.72 +/- 2.24** | **42.67 +/- 0.80** | **90.83 +/- 1.14** | **80.07 +/- 1.18** | 93.85 +/- 0.15 |
| Exphormer-2softmax | 97.83 +/- 0.24 | 74.69 +/- 0.21 | 42.25 +/- 0.75 | 90.25 +/- 0.52 | 80.00 +/- 0.87 | **94.03 +/- 0.17** |

### E.3 ABLATION STUDY: AGGREGATION

We explored alternative aggregation methods, as opposed to merely summing the two value representations described in Equation **??**. We experimented with using maximum aggregation and mean aggregation, as follows:

$$\mathbf{V} = \mathtt{max}(\mathbf{W}_{ST}\bar{\mathbf{A}}_{ST}\mathbf{V}_T, \mathbf{W}_{TS}\bar{\mathbf{A}}_{TS}\mathbf{V}_S) \qquad \mathbf{V} = \mathtt{mean}(\mathbf{W}_{ST}\bar{\mathbf{A}}_{ST}\mathbf{V}_T, \mathbf{W}_{TS}\bar{\mathbf{A}}_{TS}\mathbf{V}_S) \qquad (2)$$

Based on the superior performance for Vanilla-asym, EGT-asym, and Exphormer-asym, we select them as the baselines for this ablation study. As seen in Table 10, both the maximum and mean aggregation approaches fell short in performance when compared to the original summation method.

Table 10: Classification accuracy of Aggregation approaches

| Model | FlowGraph2 | FlowGraph3 | FlowGraph6 | Twitter3 | Twitter5 | Malnet-tiny |
|---|---|---|---|---|---|---|
| Vanilla Transformer | **95.58 +/- 0.66** | 68.22 +/- 0.55 | 39.56 +/- 1.61 | 89.12 +/- 0.43 | 77.03 +/- 1.40 | |
| Vanilla-asym | 95.50 +/- 0.20 | **69.28 +/- 0.75** | **41.42 +/- 0.78** | **91.28 +/- 0.78** | **79.42 +/- 0.48** | |
| Vanilla-asym-max | 93.62 +/- 0.87 | 68.75 +/- 1.75 | 41.31 +/- 1.67 | 89.12 +/- 0.43 | 77.06 +/- 0.81 | |
| Vanilla-asym-mean | 95.08 +/- 0.51 | 69.03 +/- 1.44 | 40.08 +/- 1.54 | 89.06 +/- 1.48 | 76.41 +/- 1.60 | |
| EGT | 95.00 +/- 1.67 | 72.06 +/- 1.16 | 42.87 +/- 0.62 | 86.49 +/- 0.73 | 73.94 +/- 1.47 | |
| EGT-asym | **96.50 +/- 0.89** | **72.39 +/- 0.55** | **42.88 +/- 0.20** | **90.37 +/- 0.29** | **82.60 +/- 0.35** | |
| EGT-asym-max | 96.50 +/- 1.47 | 69.83 +/- 2.68 | 41.22 +/- 1.23 | 85.93 +/- 2.72 | 71.59 +/- 5.74 | |
| EGT-asym-mean | 93.83 +/- 2.54 | 71.33 +/- 2.36 | 43.83 +/- 0.65 | 86.32 +/- 1.69 | 78.53 +/- 2.67 | |
| Exphormer | 96.72 +/- 0.44 | 72.81 +/- 0.38 | 41.70 +/- 0.39 | 89.76 +/- 0.30 | 72.72 +/- 1.40 | **94.02 +/- 0.21** |
| Exphormer-asym | **98.41 +/- 1.18** | **74.72 +/- 2.24** | **42.67 +/- 0.80** | **90.83 +/- 1.14** | **80.07 +/- 1.18** | 93.85 +/- 0.15 |
| Exphormer-asym-max | 98.00 +/- 0.41 | **74.72 +/- 0.96** | 42.14 +/- 0.12 | 90.82 +/- 0.35 | 80.01 +/- 0.11 | 93.62 +/- 0.08 |
| Exphormer-asym-mean | 98.16 +/- 0.59 | 74.44 +/- 0.15 | 42.42 +/- 0.59 | 90.48 +/- 0.57 | 79.63 +/- 0.43 | 93.60 +/- 0.17 |

## F COMPLEXITY AND LIMITATIONS

We designed our experiments so that we allocate the same hardware resources (GPU/CPU, amount of memory) for all experiments and also control the number of learned weight parameters to be the same for all models and for a given dataset. The recorded training timings provide a good comparison metric for empirical time complexities (i.e., given that we have constrained empirical

space complexities to be completely on par). Here are some results that show that our method is very competitive with graph transformers on this dimension: Training time for DiGT is 42s/epoch (for `FlowGraph6`) and 57s/epoch (for `Twitter3`). In comparison: for EGT (Hussain et al., 2022), the training time is 30s/epoch and 50s/epoch, respectively. Graph neural networks' training time differs significantly due to their architectures. Vanilla GCN is fast, the training time is 2s/epoch and 3s/epoch, respectively. In contrast, for PNA (Corso et al., 2020) (a GCN-based architecture), the training time is 62s/epoch and 297s/epoch.

In general, for $N$ nodes and $d$-dimensional vectors for node representations at each layer, graph transformers learn weight matrices (space complexity) with $O(N^2)$ parameters each, and graph neural networks learn weight matrices with $O(d^2)$ parameters each (with both (graph) transformers and GCN producing $N$, $d$-dimensional representations for the nodes), conducting matrix-matrix multiplications respectively of (time) complexities $O(N^2d)$ and $O(d^2N)$. Given that for the graph-level tasks we conduct, the graphs are relatively small (small $N$, comparable or smaller to $d$), this explains the generally favorable performance of our approach. On top of this, transformer architectures like ours, perform multiplications by splitting the matrix dimension $d$ into multiple heads and conducting resulting multiplications in parallel (on GPUs).

As such, scalability is a concern for most graph transformer models that leverage global self-attention (i.e., learning all-to-all node correlations as in the inspiring original transformer architecture), such as EGT, which is one of our baselines. All their experiments are conducted over collections of relatively small graphs. Our paper tries to leverage graph transformers to extract the graph structures on directed graphs in particular.