# OpenReview forum: "DIRECTIONALITY IN GRAPH TRANSFORMERS"
_ICLR.cc/2024/Conference — Submitted to ICLR 2024_

### Official Review · Reviewer_A6eP · 2023-10-28

**Soundness:** 3 good
**Presentation:** 2 fair
**Contribution:** 2 fair
**Rating:** 6
**Confidence:** 4

**Summary:**

Summary: This paper focuses on directionality of edges in graphs and introduces a new graph transformer architecture called DiGT that explicitly models edge directionality in directed graphs. The key ideas are: i) Dual node representations to capture source and target roles; ii) Directionality incorporated via asymmetric attention and dual query/key matrices, and iii) Localization via k-hop neighborhood attention.

Contributions:
- Proposes DiGT, a transformer that uses dual node encodings and directional attention to capture directionality.
- Introduces strategies to make other graph transformers directional via asymmetric/dual attention.
- New directional graph datasets where direction correlates with labels.
- Shows superior performance over GNNs and graph transformers on directional benchmarks.

**Strengths:**

Strong Points:
- Dual node encodings in DiGT elegantly capture directionality throughout the model.
- Directional attention neatly exploits query/key asymmetry for modeling direction.
- DiGT significantly outperforms baselines on directional graphs.
- Quantifies dataset directionality via entropy measure.
- New directional graph datasets enable better evaluation, although it may be biased in the context of this work.

**Weaknesses:**

Weaknesses and Questions:
- While dual encodings are powerful, they double model size. Could this be optimized?
- Although computational limitation of the proposed architecutre is discussed briefly, the runtime complexity is quadratic in number of nodes like vanilla transformers. If sparsity helps, how would the complexity differ for DiGT compared to other non-directional graph transformers?
- Are there other ways to quantify directionality of graphs besides SCC entropy?
- A closely related work "Edge Directionality Improves Learning on Heterophilic Graphs", Rossi et al., 2023, is not discussed in this work which would be important and provides (i) necessary context on novelty (ii) applicability of DirGNN on vanilla GTs.

**Questions:**

included together with weaknesses

---

> ### Author Response · Authors · 2023-11-22
>
> Thank you for your feedback. Below are our responses to the issues you've highlighted:
>
> W1. We provide the answer to this question in our general response. As noted, our model does not use more parameters, since we adjust the choices to keep it aligned with the competitive baselines.
>
> W2. The complexity of graph transformers typically depends on their structural design. Standard graph transformers, such as Vanilla Graph Transformer and DiGT, assume that all nodes are interconnected, resulting in a complexity that is quadratic relative to the number of nodes. In contrast, graph transformers employing sparse attention focus only on neighboring nodes and ignore non-existent edges. This approach reduces both computational time and memory requirements and works like graph neural networks. We will explore sparse attention techniques in the future.
>
> W3. There are no definitive measures to characterize the importance of directionality. One option is to use various directed graph metrics like distribution of in and out-degrees, directed cycles, etc. Another option is to use some spectral measures. We used the SCC entropy that combines some aspects of the directed metrics, and we also study the empirical effect of random edge flipping. We believe this is an interesting open question for the graph learning community.
>
> W4. Thank you for suggesting the Dir-GNN work, which is a directed graph neural network that focuses on incoming and outgoing edges. However, DiGT assumes that all the nodes are connected since our model is a graph-level transformer model. Dir-GNN was first published on May 17, 2023, making it concurrent to our research. We will thoroughly explore it in our future studies.

---

> > ### Comment · Reviewer_A6eP · 2023-11-22
> > **Response to rebuttal**
> >
> > Dear authors, thank you for your answers to my questions on parameters, complexity and directionality metric. Based on the merits of the paper, I retain my score.

---

### Official Review · Reviewer_yoox · 2023-10-29

**Soundness:** 2 fair
**Presentation:** 1 poor
**Contribution:** 2 fair
**Rating:** 3
**Confidence:** 4

**Summary:**

The paper proposes Directed Graph Transformer (DiGT), a global self-attention transformer specialized in encoding directed networks via dual node embeddings for source and target representations, learnable implicit adjacency information via directed attention, and k-hop neighborhood localization. For experimentation, the paper first explores the directionality of existing directed graph classification datasets by performing a random flip test, from which it finds that directionality is not a crucial factor in most datasets. Due to such limitation, the paper synthesizes two novel datasets, FlowGraph and Twitter, where the graph label is explicitly related to the edge direction pattern. Experiments on these datasets show that DiGT attains best performance across various message-passing GNN and graph Transformer baselines.

**Strengths:**

- [S1] Developing a Transformer architecture for learning directed graphs is a fairly underexplored topic, yet there is a clear demand in the community due to datasets where directionality occurs naturally.
- [S2] The overall methodology behind DiGT is well-written with great detail.

**Weaknesses:**

- [W1] **There are some questionable design choices in DiGT that seemingly contradict with the overall direction, yet are missing additional explanations.**
  - In particular, the end of Section 2 mentions how directed message-passing GNNs "suffer from convolutional inductive bias... restricted to only the given neighborhood structure", yet DiGT uses attention that is localized to the k-hop neighborhood. Doesn't this essentially downplay the advantage GTs have over message-passing GNNs?
  - Also, the end of Section 3 mentions that "DiGT uses dual node embeddings in all layers", but Equations (9) and (10) imply that dual embeddings do not remain dual throughout the encoder, but are instead merged together into a single embedding $\mathbf{Y} \in \mathbb{R}^{n \times d_p}$, then separated via linear layers $L_{VS}$ and $L_{VT}$ once every layer. Why is this the case?
- [W2] **The experimental setup is unconvincing.**
  - The paper claims that directionality does not play a significant role in existing datasets, proceeds by proposing synthetic datasets, and then performs graph classification on those networks instead. However, the first observation in Table 1 is not really comprehensive (experiments are only shown with certain model-dataset pairs), and the numbers do not necessarily overlap in standard deviation, which then leads to the question of "Are these results truly indicative of directionality not playing a significant role in these datasets?".
  - The pipeline used to generate the Twitter datasets also seems problematic, due to how the label of each graph is chosen. Specifically, Twitter5 has 5 labels, each corresponding to the perturbation rate in [0, 25, 50, 75, 100]% used to rewire or reverse each edge in the original ego-network. Then for the experiments in Table 1, randomly flipping 50% of the edges in a Twitter5 graph labeled 1 (originally perturbed by 25%) would return a graph that is equivalent to a graph labeled 3 (originally perturbed by 75%) with no edge-flipping. In essence, it is unclear whether the drop in performance shown in Table 1 is simply due to having noisy labels, rather than the dataset exhibiting significance on directionality.

**Questions:**

- [Q1] **Details on number of parameters.** Could the authors clarify the exact number of parameters used for each model in Table 2? Due to its dual attention mechanism, I suspect DiGT would use more parameters compared to other models if using the same number of layers. Having the model sizes alongside performance metrics would help clarify whether the performance gains are due to the proposed mechanisms, and not due to having more parameters.
- [Q2] **Missing results for Malnet-tiny of Tables 2 and 3.** Are these all blank due to out-of-memory issues? If so, it would be better to fill them in with OOM.

---

> ### Author Response · Authors · 2023-11-22
>
> We want to thank you for your thoughtful ideas. Here is our response to the concerns you raised regarding potential weaknesses:
>
> W1. We choose to retain the k-hop constraint, since it helps somewhat, as shown in Table 11 in the appendix. We replicate the last two rows of that table here:
>
> | Model              | Flow2 | Flow3 | Flow6 | Twitter3 | Twitter5 |
> |--------------------|-------|-------|-------|----------|----------|
> |DiGT-DA (no k-hops) | 96.52 | 74.33 | 46.22 | 91.31    | 85.46    |
> |DiGT                | 97.42 | 74.55 | 46.80 | 91.67    | 85.94    |
>
> As we can see the use of k-hops confers a slight benefit. But, one critical difference from GCNs should be noted. The GCNs are always constrained by the direct neighbors which add a very strict inductive bias. On the other hand, when we used k-hops, we treat the entire set of k-hop neighbors as the context, i.e., any node can directly communicate with any other node *within* the k-hops, which is a much less restrictive "bias", and lets us tradeoff the global attention with somewhat localized attention.
>
> When we say that we use dual encodings throughout, it does not imply that there is no exchange between these source and target embeddings. In fact, it is clearly of benefit to allow information to flow from one to the other, enabling them to mutually exchange information.
> We do not want to  reduce the system to a mere duplication of single-encoding attention, so it is not clear whether the reviewer is suggesting that keeping them independent is a better choice?
>
> W2. We refer the reviewer to Tables 6 and 7 in the appendix, that already does a comprehensive comparison of all the flipping experiments to show the importance of directionality.
>
> In these experiments, we implement a strategy of randomly shuffling the dataset at each training step. This approach means that the dataset appears different in each of the 100 epochs during model training. As a result, a slight decrease in model performance is anticipated, given that the model encounters a 'new' dataset each time. However, for datasets like MNIST, CIFAR10, and Ogbg-Code2, we observed only about a 1% drop in model accuracy. This finding suggests that directional information is considerably less significant, and potentially negligible, compared to the node or edge features.
>
> Since the study of flipping edges is an independent experiment, we establish a standard pipeline applicable to all datasets. In the case of the Twitter dataset we first generate Twitter datasets for all the experiments. Then, for the random flip study, we randomly flip the edges for each step. The changes in the accuracy highlight the impact of edge directionality in this dataset, though there may also be an effect of class label confusion. Please note that, whatever may be the final effect, DiGT displays superior performance compared to all other (undirected) models, which leads further credence to the important role of directionality in the Twitter dataset.
>
>
> Q1. We provide the answer to this question in our general response. As noted, our model does not use more parameters, since we adjust the choices to keep it aligned with the competitive baselines.
>
> Q2. Thank you for your suggestions. We will mark "OOM" on these tables.

---

> ### Comment · Reviewer_yoox · 2023-11-23
> **Response to Author Rebuttal by Reviewer yoox**
>
> Thank you authors for your commitment into providing additional clarifications. Many of my concerns have been addressed, but I still have two concerns remaining:
>   - **Interactions between dual embeddings.** I agree that exchanging information across source and target embeddings would help in performance. However, due to commutativity of vector addition, adding the two embeddings and processing the result through two separate linear layers to obtain dual embeddings for the next layer (Equations 9 and 10) loses information on the ordering of the two embeddings, and thus we can no longer tell which embedding corresponds to the source or target. One simple way to exchange information while preserving the source/target ordering would be to fuse the embeddings via  $S' = S+ L_{VS}(T)$ (similar to feature fusion in [A]), instead of $S' = L_{VS}(S+T)$ which is used in the presented work. Did I miss something that makes this commutativity a nonissue?
>   - **Experiments under Randomized Directionality.** My previous concern was that Table 1 only shows results from certain model-dataset pairs, and does not seem conclusive to say that "directionality does not play significant role in these datasets." While the authors have referred to Tables 6 and 7 in response, I am guessing the authors meant Tables 3 and 4 since Tables 6 and 7 pertain to another experiment. Nonetheless, these results still do not cover all model-dataset settings and the numbers still are not really convincing.
>
> Because of these remaining concerns, I will retain my score.
>
> [A] Zhu et al., Graph Geometry Interaction Learning. NeurIPS 2020.

---

### Official Review · Reviewer_r1B3 · 2023-10-30

**Soundness:** 2 fair
**Presentation:** 3 good
**Contribution:** 3 good
**Rating:** 5
**Confidence:** 4

**Summary:**

The paper proposes a directional graph transformer that utilizes dual encodings to represent the different roles of source or target of each connected node pair. The dual encodings are acquired through the utilization of latent adjacency information, which is extracted using the directional attention module localized with k-hop neighborhood information. Additionally, the paper introduces alternative methods for incorporating directionality within the Transformer architecture. In the experimental study, the paper examines the role of directionality in current datasets, and proposes two new directional graph datasets. By conducting a comparison on directional graph datasets, the authors demonstrate that their approach achieves state-of-the-art results.

**Strengths:**

1. The work introduces a method that does not fully rely on the explicit directed graph structure, which allows the central node to receive extra information from non-neighbors. At the same time, the positional encodings and the k-hop localization ensure that the attention scores do not deviate too much from the original data structure.

2. The proposed model outperforms GT and GNN alternatives on five reported datasets. It also surpasses the other directional GTs on three out of five datasets.

3. The paper provides two new datasets that offer new benchmarks to evaluate directional graph modeling.

**Weaknesses:**

1. How the learnt attention scores are related to the original graph structure or the edge directions is not fully revealed (probably can be done by visualization and comparison).

2. Some designs are not well justified by either theoretical or empirical evidence. For example, the design of the positional encoding. No illustration, for instance, unidirectional counterparts or theoretical justification, has been provided about why the design is suitable for the directional situation.

3. The computational complexity of the proposed model appears high. It may have scalability issues when applied to large datasets.

**Questions:**

1. Would it be possible to include some node-level experiments to further examine the capability of the model? There are a few commonly used benchmark datasets with directed graphs for node level tasks, e.g., Actor. Are such graph datasets too large for the proposed model to handle?

2. Based on the recent literature, [1] also presents a positional encoding design tailored for directed graph Transformers. It would be valuable to see a comparative evaluation of your proposed model with this work.

[1] Geisler, Simon, Yujia Li, Daniel J. Mankowitz, Ali Taylan Cemgil, Stephan Günnemann, and Cosmin Paduraru. "Transformers meet directed graphs." In International Conference on Machine Learning, pp. 11144-11172. PMLR, 2023.

---

> ### Author Response · Authors · 2023-11-22
>
> We appreciate your positive feedback. Here are our answers to the points raised:
>
> W1. Visualizing the attention scores would be interesting, something we can explore for the future.
>
> W2. We used SVD-based positional encodings (PEs) since the left and right singular values/vectors can be used very naturally for directed graphs, unlike the Graph Laplacian, which is symmetric. For a comparison with Magnetic Laplacian PEs, see Q2 below.
>
> W3. We have included the number of parameters used by our model and others in the general response above. Thus, our method is no more and no less scalable than other (dense) transformer based models.
>
> Q1. We ran our model on the WebKB and the Telegram datasets. WebKB contains Cornell, Wisconsin, and Texas datasets. The table below reports on the accuracy of DiGT vs other baselines; we see clear improvements using DiGT (we do not include std-dev for clarity, but we will include them in our revised paper). Nevertheless, transformer-based models work only for medium sized datasets; for large number of nodes all (dense) attention methods will have scalability issues.
>
>
> | Datasets    | Cornell | Wisconsin | Texas | Telegram |
> |-------------|---------|-----------|-------|----------|
> | DGCN [ii]   | 65.13   | 71.08     | 64.90 | 85.77    |
> | DiGCN [i]   | 45.68   | 50.54     | 53.73 | 65.96    |
> | DiGCNIB [i] | 41.08   | 50.81     | 56.47 | 56.35    |
> | EGT         | 71.43   | 73.59     | 81.18 | 80.00    |
> | DiGT        | 71.62   | 74.05     | 83.73 | 90.38    |
>
> [i] Z. Tong, Y. Liang, C. Sun, X. Li, D. Rosenblum, and A. Lim. Digraph inception convolutional networks. Advances in Neural Information Processing Systems, 33, 2020.
>
> [ii] Z. Tong, Y. Liang, C. Sun, D. S. Rosenblum, and A. Lim. Directed graph convolutional network. arXiv preprint arXiv:2004.13970, 202
>
> Q2. Thanks for suggesting the use of the Magnetic Laplacian PEs, which are indeed even better than SVD based positional encodings, as shown in the table below (we will include these new results in Table 2 in our updated paper; we do not show the std-dev values below for clarity, but we will include those as well):
>
>
> | Datasets | Flow2 | Flow3 | Flow6 | Twitter3 | Twitter5 |
> |----------|-------|-------|-------|----------|----------|
> | DiGT-SVD | 97.42 | 74.55 | 46.80 | 91.67    | 85.94    |
> | DiGT-Mag | 97.00 | 74.92 | 47.49 | 93.34    | 86.61    |
> Here DiGT-Mag is with the magnetic Laplacian position encodings.

---

> > ### Comment · Reviewer_r1B3 · 2023-11-22
> > **Thank you for the response**
> >
> > Unfortunately, most of my concerns are not addressed (detailed below). I will keep the current rating.
> >
> > W1: Visualization of attention scores is yet to be provided. It is important to provide some experimental results to prove that DiGT does learn some directional information within graphs.
> >
> > W2: The relationship between the left/right singular values/vectors and the direction of edges is not revealed.
> >
> > W3&Q1: The response somehow reveals the scalability issue of DiGT. The WebKB datasets are quite small in node-level tasks. For example, Cornell only contains 183 nodes. The suggested dataset, Actor, is also considered as a small graph in node-level tasks, which contains 7600 nodes. However, the authors did not provide experimental results on this dataset.
> >
> > Q2: Magnetic Laplacian position encodings seem to outperform DiGT-SVD.

---

### Official Review · Reviewer_1Uzt · 2023-10-31

**Soundness:** 2 fair
**Presentation:** 2 fair
**Contribution:** 2 fair
**Rating:** 3
**Confidence:** 4

**Summary:**

This paper introduces Directed Graph Transformer (DiGT), a novel graph transformer architecture that effectively addresses the challenge of analyzing directed graphs. DiGT incorporates edge direction and graph connectivity as integral elements within the Transformer framework, enabling it to dynamically learn dual node encodings that capture the directionality of edges. Experimental results demonstrate that DiGT significantly outperforms state-of-the-art graph neural networks and graph transformers in directed graph classification tasks, particularly when edge directionality is a crucial aspect of the data.

**Strengths:**

1) Exploring the utilization of graph transformers to encode the directed information within graphs is a promising avenue of research.
2) DiGT, as an approach that leverages a graph transformer to encode directedness, shows promise, and the experimental results provide evidence of its effectiveness to a certain extent.

**Weaknesses:**

1) The three modules of DiGT are all based on heuristic methods, lacking some theoretical explanations and insights.
2) DiGT contains many linear layers and learnable parameters, making it quite complex. While the author briefly describes the complexity of DiGT in Appendix F, I recommend conducting a more detailed theoretical analysis and empirical validation.
3) In the experiments, the author mentions abandoning the use of some available datasets because they think the directionality of these datasets is unimportant. I don't entirely agree with this viewpoint. I think that even if directionality is not crucial, if a model can encode directionality, it should still yield some benefits.
4) Some important baselines were not compared in the experiments. For example, the method proposed in the paper "Transformers Meet Directed Graphs [1]" achieved the SOTA result on Ogbg-Code2.

[1] Geisler, Simon, et al. "Transformers meet directed graphs." _International Conference on Machine Learning_. PMLR, 2023.

**Questions:**

Please refer to the aforementioned weaknesses.

---

> ### Author Response · Authors · 2023-11-22
>
> We want to thank you for your thoughtful comments. Below are our responses:
>
> W1. Our DiGT model is inspired by the well known HITS approach for directed graphs using notion of authority and hub scores (which, are eigenvectors of the $A^T A$ and $A A^T$ matrices). Our key innovation is the use of source/target vectors (embeddings) with a learnable adjacency matrix in the HITS approach, and thus it has a solid theoretical foundation.
>
> W2. We not only suggest a novel learnable formulation for the directed embeddings, but we also study alternative formulations for incorporating directionality, which we study empirically. Thus, our work is not just a set of ad-hoc methods, but rather a systematic study of how direction can be added effectively within transformer models.
>
> W3. We disagree. If a dataset has no directionality, then how or why will a directional transformer help? For example, if a graph is undirected, then its adjacency matrix A, will be symmetric, and the notion of hub or authority collapse -- $A^TA$ and $A A^T$ will be the same.
>
> W4. Please note that directionality is not significant in the original ogb-code2 dataset, as we have shown in Table 1. Furthermore, please note that Geisler et al preprocess and modify the original graph to get competitive results, but we consider the original graph, which has little directionality.

---

> > ### Comment · Reviewer_1Uzt · 2023-11-22
> > **Re**
> >
> > Thank you for the response. After reading the rebuttal and other reviews, I will keep my score.

---

### Author Response · Authors · 2023-11-22

Since several reviewers had questions about the number of parameters for our DiGT model versus the other baseline methods, we show in the table below that our model has a comparable number of parameters:


|Models      | DiGT  | EGT    | Exphormer |
|------------|-------|--------|-----------|
|#Parameters | 91446 | 115410 | 102246    |
|#Layers     | 4     | 4      | 4         |
|#Heads      | 8     | 8      | 4         |
|Hidden dim  | 32    | 48     | 32        |


The numbers are shown for the Flow2 dataset, with similar values for other datasets and models. For all experiments, we consistently use around 100K parameters (for malnet-tiny Exphormer uses 400K parameters).

---

### Meta-Review · Area_Chair_oJRh · 2023-12-09

**Metareview:**

This paper proposes Directed Graph Transformer (DiGT), a new graph transformer architecture for analyzing directed graphs. DiGT incorporates edge direction and graph connectivity as integral elements in Transformer, enabling to dynamically learn dual node encodings that capture the directionality of edges. Experimental results show that DiGT improves over graph neural networks and graph transformers in directed graph classification tasks. As pointed out by reviewers, the paper needs to compare with necessary baselines; some of the modeling designs are not well justified by either theoretical or empirical evidenc; and the computational cost (e.g., due to dual encoder) of the proposed model seems high.

**Justification For Why Not Higher Score:**

the paper needs to compare with necessary baselines; some of the modeling designs are not well justified by either theoretical or empirical evidenc; and the computational cost (e.g., due to dual encoder) of the proposed model seems high.

**Justification For Why Not Lower Score:**

NA

---

### Decision · Program_Chairs · 2024-01-16

Reject